# Psychosocial Determinants of Adolescents’ Cyberbullying Involvement—The Role of Body Satisfaction

**DOI:** 10.3390/ijerph19031292

**Published:** 2022-01-24

**Authors:** Marta Malinowska-Cieślik, Anna Dzielska, Anna Oblacińska

**Affiliations:** 1Department of Environmental Health, Faculty of Health Sciences, Medical College, Jagiellonian University, 31-066 Krakow, Poland; 2Department of Child and Adolescent Health, Institute of Mother and Child, 01-211 Warsaw, Poland; anna.dzielska@imid.med.pl (A.D.); anna.oblacinska@imid.med.pl (A.O.)

**Keywords:** adolescents, cyberbullying, body image, social support, psychosocial factors

## Abstract

Background: Although the relationship between adolescents’ body image and cyberviolence has been proved, little is known about the place of body image among other psychosocial determinants. The study aimed to assess the relationship between body satisfaction and cyberbullying in the context of other psychosocial factors. Methods: We used data of 5817 adolescents (47.6% boys, mean age = 15.43) from the survey conducted in 2018 in Poland as a part of the Health Behaviour in School-Aged Children. Four groups of cyberbullying involvement were defined: only bullies, only victims, both bullies and victims, and not involved. Body satisfaction and ten other independent variables were classified as sociodemographic, socioeconomic, individual and social factors. A series of multinomial logistic regression models were estimated and compared using the pseudo R-sq Nagelkerke coefficient. Results: Although family support seemed to be the most protective, the findings have proved that body satisfaction reduces significantly the risk of cyberbullying. The relationship was more pronounced in victims and bully-victims. A slightly stronger protective effect of body satisfaction has been observed in boys. Conclusions: The strengthening of body image may be an effective measure to prevent adolescents’ cyberbullying, as well as to bring about improvements in social support, connections to others, and school performance.

## 1. Introduction

The development of modern technologies has an impact on the emergence of new threats in the lives of adolescents. The internet and social media are becoming a space of both positive and negative social experiences related to the mental health and well-being of teenagers [1]. One of these experiences is cyberbullying, which can be defined as an intentional behavior aimed at harming another person or persons by means of electronic devices, such as computers, mobile phones and others that are perceived by the victim as an aversion [2]. Although some researchers say that cyberbullying is a new form of peer violence, others emphasize that the co-occurrence of these phenomena does not necessarily mean conceptual convergence. The roles of bullies and victims may overlap and appear in different contexts, both face to face and in cyberspace, where traditional bullying victims may change roles and become cyberbullying perpetrators [3]. Experts believe that consequences of cybervictimization may be more severe than those of traditional victimization [4]. Studies show various negative impacts of cyberbullying on adolescents’ mental health, like suicidal thinking, depressive symptoms, loneliness, frustration, and sadness, as well as difficulties in school performance, in learning and low school achievements [5,6,7,8].

Research examining bullying and cyberbullying has primarily focused on two categories of involvement: those who are victims and those who are perpetrators [9]. However, the peer interpersonal violence and cyberviolence involves not only bullies and victims, but also bully-victims, who both bully and are bullied by others [10,11]. It appears that adverse mental health outcomes due to bullying in adolescence most severely impact on bully-victims [12]. These studies explore the factors that contribute to the involvement of youth in cyberbullying by distinguishing them according to four categories of involvement: victims, bullies, bully-victims, and not involved.

Furthermore, studies on adolescents’ cyberbullying show that the most common reasons for being persecuted in cyberspace are related to appearance, weight-related teasing, style of clothing or type of silhouette [13]. Girls often receive comments about being fat, while among boys, it is common to receive comments about looking or seeming ‘gay’ [14]. The prevalence of cyberbullying is related to a negative perception of one’s own appearance and low body esteem, and this causes mental health problems and decreases well-being [15]. Adolescents who perceive their body negatively are more frequently involved in cyberbullying than their peers who are satisfied with their appearance [16,17,18]. They may manifest body dissatisfaction by displaying aggressive or passive attitudes towards their peers, which increase the risk of bullying or being a victim as a result of a disturbed relationship [19]. Victims of cyberbullying report poorer body esteem than nonvictims. Studies analyzing the associations of cyberbullying with body image moderated by social support and other individual factors, indicate that low body esteem and dissatisfaction predict victimization [13,17,18,20].

During adolescence, dramatic changes in physical, cognitive, emotional and social development occur in individuals’ transition [21]. Therefore, in the literature, certain individual risk factors related to adolescents’ cyberbullying have received special attention, including demographic characteristics, such as age and gender, beliefs and attributions towards self, and body-related self-esteem, social self-esteem, loneliness, problem-solving skills, and school performance and satisfaction [8,15,22,23,24]. Body dissatisfaction has been found to be a significant reason for cyberbullying among adolescents. The mediating role of body image, and the relationship between cybervictimization and body dissatisfaction among adolescents has been proved [17,18,20,25]. Moreover, studies which review other psychosocial correlates with cyberbullying highlight the importance of empathy, moral engagement and commitment to respecting others [26,27].

Besides the number of individual factors related to adolescents’ cyberbullying, the social ecological perspective draws attention to a wider protective context [28,29]. The social microsystem includes social support, such as relationships with family and peers, school social climate and community social capital. The social support from family and friends offers the strongest protection against cyberbullying [30,31,32]. The review study of López-Castro and Priege [33] found evidence that the most consistent family variables influencing cyberbullying are family communication and the quality of the family relationships.

A unique source of data on sociodemographic, socioeconomic, and psychosocial determinants of adolescents’ well-being and health is the Health Behaviour in School-Aged Children (HBSC) the WHO Regional Office for Europe collaborative cross-national survey. In a combined international sample from the latest HBSC study conducted in 2018 in 45 European countries and regions and Canada, the prevalence of cyberbullying is differentiated according to gender and age. Among boys, the percentage of perpetrators and victims of cyberviolence are similar (12%). Meanwhile, girls are more likely to be exposed to cyberbullying (14%) than boys and are less likely to be the perpetrators (8%). In perpetration, gender differences increase with age, and in victimization, gender differences are highest at age 13 in most countries, including Poland. The prevalence of cyberbullying varies greatly among the HBSC countries, and the differences for both forms of cyberbullying involvement reach almost 30%. In international rankings, Polish teenagers occupy a very unfavorable position, being in the top ten countries with the highest percentage of young people experiencing cyberviolence, as well as those who commit online aggression. Moreover, 15-years-olds have one of the highest rates both of cyberviolence perpetration and victimization—the fourth position in the ranking among all HBSC countries [34]. Considering the body related self-esteem, in international rankings based on the findings of the recent HBSC study, Polish youths take first and second position, being in the top five countries with the highest percentage of young people who are not satisfied with their body. Moreover, the perception of the body among Polish adolescents worsens with age, both in girls and in boys [34].

Despite the extensive literature about psychosocial determinants of adolescents’ cyberbullying victimization, we found a lack of a more comprehensive analysis of the role of body satisfaction (BS) with various types of cyberbullying involvement. Therefore, we have set the aim of this study to explore and assess the relationship between body image and adolescents’ cyberbullying involvement as only bully, only victim and bully-victim in the context of other psychosocial predictors.

We hypothesized that body satisfaction would decrease the risk of adolescents’ cyberbullying involvement. Furthermore, we hypothesized that the sociodemographic, individual and social factors would moderate the influence of body image on cyberbullying involvement among adolescents. We have posited four research questions.

*RQ1:* How often do Polish adolescents have experience of various types of cyberbullying involvement according to their sociodemographic characteristics?

*RQ2:* What are the differences between the psychosocial determinants of being a cyberbully, a cybervictim or both cyberbully-victim?

*RQ3:* In what group, which type of cyberbullying involvement is the association with body satisfaction most clearly evident?

*RQ4:* To what extent do sociodemographic, socioeconomic, and other psychosocial factors moderate the relationship between body satisfaction and cyberbullying involvement?

## 2. Materials and Methods

### 2.1. Sample and Procedure

The present study used data from a cross-sectional survey implemented in Poland as part of the HBSC conducted in 2017/2018, using a standardized and validated questionnaire [35]. Translation and back translation of the scales included in the questionnaire were adopted. The translation procedure of the questionnaire followed an international HBSC survey protocol [36]. All questions were translated from the original English version into the Polish. Thereafter the Polish translation was back translated into English by an independent translator and submitted to the HBSC Translation Hub. The translation was reviewed, accepted or modified according to the reviewer’s comments.

The study design employed stratified cluster sampling with classes within schools as the primary sampling unit. Schools were randomly selected, and individual classes within these schools were subsequently randomly included. Self-completion questionnaires were administered in the classroom. Consent was obtained from school administrators, parents or caregivers and students. The principals of the schools were informed about the aim of survey and their supervisory units. External trained interviewers were responsible for data collection. Adolescents’ participation was anonymous and voluntary, and no incentives were offered for participation. The study received approval from the Bioethical Commission operating at the Institute of Mother and Child in Warsaw (No. 17/2017 with Annex 1, dated 30 March 2017). 

The study sample was composed of 5838 adolescents (52.4% female). Data analysis was based mainly on answers from 5817 pupils, who responded to the cyberbullying questions, while 21 were missing data. The data were collected in three age groups 13, 15 and 17 years old: 36.7% were 13 years old, 34.1% were 15 years old, 29.2% were 17 years old, and the mean age was 15.43 years (SD = 1.73). Among respondents 75.2% lived in an intact family, 16.5% in a single-parent family, 5.3% in a stepfamily, and 3.1% in a family without parents. The study sample characteristics are described further in the tables. 

### 2.2. Measures

#### 2.2.1. Cyberbullying

Cyberbullying involvement was the main outcome variable. The questionnaire included one question about perpetrating and one about being the victim, using an item modified and adapted from the revised Olweus Bully/Victim Questionnaire [37,38]. Participants answered two questions to indicate whether they had experienced or perpetrated cyberbullying in the two months prior to the administration of the survey with five response categories ranging from ‘I have not been bullied in this way in the past two months’ to ‘several times a week’.

On the basis of these two questions, new variables were created, which illustrate the type of cyberbullying experience. The study sample was thus divided into four separate groups regarding cyberbullying involvement: only perpetrators of cyberbullying, only victims of cyberbullying, both perpetrators and victims of cyberbullying, and those not involved in cyberbullying as the reference group.

#### 2.2.2. Family Structure and Socioeconomic Status 

Students reported on a number of sociodemographic characteristics including gender and age. Students were asked about family structure, who they live with in the home in which they live all or most of the time. Response options were ‘mother’, ‘father’, ‘stepmother (or father’s girlfriend/partner)’, ‘stepfather (or mother’s boyfriend/partner)’, ‘someone or somewhere else (e.g., siblings, grandparents)’ and ‘foster or children’s home’. Four categories of family structure were then derived: intact family, single-parent family, stepfamily and family without parents. Socioeconomic status (SES) was assessed using the HBSC Family Affluence Scale (FAS III), which includes six items that measure material assets in the home, such as number of vehicles, bedroom sharing, computer ownership, bathrooms at home, dishwashers at home, and family vacations [39,40]. The item scores were summed with a score from 0 to 13 and categorized into three SES groups: low (0–5), average (6–9) and high (10–13). In the correlation analyses family affluence (FA) variable was used as continuous variable. A reliability index (Cronbach’s alpha = 0.561) was less than 0.7. Even if Cronbach’s alpha for FAS III was less than 0.7 with PCA only 33.5%, this tool is recommended for use in HBSC study. The six-item scale was estimated with Samejima’s graded response model and tested for differential item functioning by country (also Poland) under the last FAS III scale validation study [40]. This study the test-retest reliability for Poland was r = 0.91 and the FAS scale correlated with the family income reported by parents with the Eta^2^ close to 0.30.

The family social status (FSS) was measured by respondents’ subjective assessment on the 11-rung ladder from 0 to 10 points ranking the family social status [41]. The item scores were categorized into three groups: low (0–5), average (6–8) and high (9–10). 

#### 2.2.3. Individual Factors

The body satisfaction (BS) was measured by the Body Image Subscale (BIS), which consists of six scale items. BIS is an element of the Body Investment Scale modified version [42]. This scale includes six items, and each item is rated on a five-point Likert scale. Respondents were asked to answer statements about their feelings related to their body and physical appearance. The item scores were summed with a score from 0 to 24 and categorized into three levels: low (0–12), medium (13–20) and high (21–24). The internal consistency of the BIS was high (Cronbach’s alpha = 0.901).

Social self-efficacy (SSE) was measured by subscales of the Self-Efficacy Questionnaire for Children (SEQ-C) by Muris [43]. This scale includes eight items, which are rated on a five-point scale from 0 (not at all) to 4 (very well). The item scores were summed with a score from 0 to 32 and categorized according to three levels: low (0–16), average (17–25) and high (26–32). The internal consistency of the scale was good (Cronbach’s alpha = 0.842).

School achievements (SA) were measured by self-rating one’s own academic status compared to classmates via an 11-rung ladder from 0 points (the worst) to 10 points (the best academic achievements) [44]. The item scores were categorized into three groups: low (0–4), average (5–7) and high (8–10).

#### 2.2.4. Social Factors

Given the possible moderating role of social support in the associations between adolescents’ body image and cyberbullying, the family and peer support subscales of the Multidimensional Scale of Perceived Social Support (MSPSS) [45] were applied. Family support was measured using the family relation items, which include statements describing the degree of help, emotional support, communication and assistance in decision-making in the family, and respondents were asked to rate the four statements on a five-point Likert scale. Family support was assessed using the scale response descriptors, where total score ranged from 0 to 24, and 0–12 were considered as weak; a score of 13–22 was average, and a total score of 23–24 was categorized as strong support. The family support scales have shown good internal consistency (Cronbach’s alpha = 0.936). Peer support was measured using four items, which describe the degree of help from friends, ability to count on them, communication of feelings, and of problems with friends on a five-point Likert scale. A variable was computed to calculate a mean score for all participants who answered the four items within the above scale. Total score ranged from 0 to 24, was assessed as weak (0–9), average (10–19) and strong (20–24). These measures have shown good internal consistency (Cronbach’s alpha = 0.895).

Connections to others is one of the spiritual health scale domains. This scale was tested in 2013 in Canada and Scotland and then explored in six HBSC countries [46]. The connections to others subscale contains three items: two from the original tool and one added to the HBSC 2017/18 protocol. Students had to identify how important it is to ‘be kind to other people’, ‘be forgiving of others’ and ‘show respect for other people’. Response categories for all items ranged from 0, ‘not at all important’, to 4, ‘very important’. The item scores were summed with a score from 0 to 12, with three levels: weak (0–8), average (9–10) and strong (11–12). Internal consistency between these items was good (Cronbach’s alpha = 0.863).

### 2.3. Statistical Analysis

The psychometric properties of all scales described above are presented as unpublished electronic material (Table A1), while mean values and standard deviations are provided together. Cronbach’s alpha coefficients were estimated, with values above 0.8 indicating good internal consistency [47]. Only the FA has a reliability index of less than 0.7. However, it is a widely accepted tool implemented by the HBSC study network. All six items of FAS are homogenous. The relationship between the type of experiences with cyberbullying and 11 variables grouped in three thematic blocks was analyzed, such as sociodemographic factors (gender, age, family structure, FA, FSS), individual factors (BS, SSE, SA), and social factors (family and peer support, connections to others). In the first step, the analysis was carried out for categorized variables using the chi-squared Pearson tests. The associations of adolescents’ involvement in cyberbullying with sociodemographic, individual and social factors were examined. In the second step, the correlation between quasicontinuous scales was investigated using Spearman’s rho coefficient. In the third step, a series of logistic multinomial regression models were estimated, including successive blocks of factors as independent variables. Regression analysis was conducted to determine the predictive value of each variable included in the model. The dependent variable took four values, corresponding to cyberbullying involvement as a bully, victim, bully-victim and not involved. The reference category included adolescents never involved in cyberbullying. This method allows to predict nominal outcome and to assess in one model the risk of being only the perpetrator, only the victim, or both the perpetrator and the victim of cyberbullying. Subsequent models were compared using the pseudo R-sq Nagelkerke goodness-of-fit coefficient. Attention was paid to how the conclusions concerning the influence of particular factors change in the increasingly extended models. The main focus was, according to the hypotheses, to present the influence of BS on cyberbullying among adolescents, adjusted for individual and social cofactors. The effect of excluding each of the 11 factors was confirmed by a log-likelihood test. The data were analyzed using IBM SPSS statistics software package, version 23 (IBM Corp., Armonk, NY, USA) and a NOMREG procedure. Statistical significance was established a priori at *p* < 0.05.

## 3. Results

### 3.1. Prevalence of Cyberbullying in Total and by Sociodemographic Factors

Among the study group, 75% were not involved, 7.1% were only bullies, 7.8% only victims, and 10.1% were bully-victims. The percentages of pupils with different bullying experiences according to four categories were analyzed in groups with 11 variables found to be predictors of the cyberbullying. The variables were classified into three blocks: sociodemographic, individual (including BS) and social factors. Table 1, Table 2 and Table 3 summarize the descriptive statistics. The total number of the study group included 5817 adolescents. However, there are different sums in variables due to missing data.

Differences in cyberbullying involvement by gender, age and family structure were evident, and rates varied substantially in these groups, whereas FA and FSS did not impact involvement in cyberbullying (Table 1). The percentage of girls without any cyberbullying experience was 5.1% higher than boys (chi-sq (3, N = 5817) = 77.49, *p* < 0.001). With respect to bullies and bully-victims, rates were higher among boys compared to girls, while victims were more prevalent among girls. The highest percentage of students who were not involved in cyberbullying was observed among the oldest students, while the lowest rates occurred among 15-year-olds (chi-sq (6, N = 5817) = 38.26, *p* < 0.001). This age group represented the highest tendency to be a perpetrator as well as a bully-victim. The highest percentage of victims was observed among the youngest students. Considering family structure, the difference in cyberbullying involvement was significant (chi-sq (9, N = 5817) = 28.94, *p* < 0.001). The highest percentage of adolescents not involved in cyberbullying was observed among students from the intact families, while the lowest rates occurred among students growing up without parents. Comparing three types of cyberbullying involvement by family structure, the highest difference was observed in bully-victims. The lowest percentage of bully-victims was recorded among teenagers living with both biological parents, and in subsequent groups, involvement in the mixed cyberbullying model was significantly higher. The highest percentage of bully-victims was observed among pupils living in families without parents.

### 3.2. Prevalence of Cyberbullying by Psychosocial Factors

The analysis of the relationship between cyberbullying and individual factors (Table 2) showed significant differences in the percentages of students who did not engage in cyberbullying and those who did, depending on BS (chi-sq (6, N = 5686) = 89.65, *p* < 0.001), SSE (chi-sq (6, N = 5636) = 25.54, *p* < 0.001), and school achievements (chi-sq (6, N = 5772) = 31.01, *p* < 0.001). The percentage of teenagers who did not experience any form of cyberbullying increased with an improvement in BS. The percentage of perpetrators was highest among students with high BS compared to those less accepting of their body. The greatest differences depending on the BS were recorded in victims and bully-victims. More than twice as many victims occurred among adolescents not satisfied with their body compared to those who were satisfied. Perpetration and victimization also occurred most often among adolescents dissatisfied with their body in contrast to among those with higher BS. A similar pattern to that in the case of body image was also observed regarding SSE. A higher percentage of students with stronger SSE were not involved in cyberbullying compared to those with low SSE. Rates of victims and bully-victims were higher in students with low SSE. In bullies an opposite direction was observed. Students with high SSE bullied more frequently than those with lower levels of SSE. The percentage of students who were not involved in cyberbullying among those with very good school results, was almost 8.9% higher than among those with the lowest achievements. As school achievements improved, the percentage of students who were perpetrators, victims and bully-victims decreased.

Table 3 shows the percentages of adolescents with different experiences taking social factors into account. A relationship has been confirmed for all three variables: family support (chi-sq (6, N = 5751) = 90.86, *p* < 0.001), peer support (chi-sq (6, N = 5776) = 49.38, *p* < 0.001) and connections to others (chi-sq (df = 6, N = 5787) = 248.38, *p* < 0.001). Among students with strong family support, the proportion of those who were not involved in cyberbullying was 14.6% higher than among those with low support. Rates of bullies, victims or bully-victims were higher in adolescents perceiving weak support compared to those with strong family support, with the most notable difference in the bully-victims group (7.8%). A similar pattern was observed regarding peer support. In students with strong peer support, the proportion of those who were not involved in cyberbullying was 9.3% higher than among those with weak support. Comparing three groups of cyberbullying, the highest difference was observed in the bully-victim group, and 5.5% higher among students with weak support than among those with strong peer support. In adolescents with strong connections to others, the percentage of those not involved in cyberbullying was 18.6% higher than among those with low level. The proportion of bullies and bully-victims was higher among students with weak connections to others compared to those with strong relationships, and the differences between this groups were 7.1% and 13.9% respectively. In victims, an opposite direction was observed. The students with strong connections to others were more frequently victimized than those with weaker relationships.

### 3.3. Correlation between Psychosocial Determinants of Cyberbullying 

Table 4 shows the correlation of eight factors that may potentially affect being involved in cyberbullying among teenagers. In only one case, connection to others and FA, was no correlation detected (rho = 0.004, *p* = 0.778). In most cases, Spearman’s rho coefficients were positive and differed significantly from zero. In one case, BS and FA, was the relationship significant, but the rho value indicated a negligible relationship (rho = 0.040; *p* = 0.003). The strongest correlation was obtained for SSE with peer support (rho = 0.457; *p* < 0.001). Due to the value of the coefficient, statistically significant correlations were recorded for the relationship of BS with family support (rho = 0.304; *p* < 0.001), and family social status with family support (rho = 0.341; *p* < 0.001).

We checked the assumptions regarding multicollinearity, to ensure the regression-type analysis, that the models were valid [48,49]. The values of the correlation coefficients between independent variables used in our study were quite low. According to general assumptions it allowed the regression analyses to be performed, and to estimate the models.

### 3.4. Multifactorial Logistic Regression

A series of multinomial logistic regression models were estimated and compared on the basis of the Nagelkerke’s R-sq goodness-of-fit statistics. In subsequent models, the set of factors that may influence the risk of various types of cyberbullying involvement has been gradually increased. The exact results of intermediate and final model estimation are presented in Table 5 and Table 6. In the simplest model including only sociodemographic factors (gender, age and family structure), Nagelkerke’s R-sq was equal to 0.030. By adding BS, a significant improvement in model quality was achieved (Nagelkerke’s R-sq = 0.056). Small improvement has been achieved by adding the other two variables relating to individual factors, SSE and school achievements (Nagelkerke’s R-sq = 0.062). The addition of two factors reflecting the FA and family social status only slightly changed the goodness-of-fit statistics (Nagelkerke’s R-sq = 0.065). The model obtained by adding two factors related to social support from family and peers had much better predictive value (Nagelkerke’s R-sq = 0.078). In the final model when the connections to others had been added, the R-sq increased to 0.111. Comparison of the above five models shows that cyberbullying involvement can be explained to a relatively large extent by weak connections to others, low BS and weak family support. 

In Table A2, provided as supplementary electronic material, the results of the log-likelihood test comparing full model with reduced models is shown. It represents the effect of excluding each factor and confirms the highest impact of three predictors mentioned above, connections to others, family support and BS. The effect of SA, SSE and FA appeared to be marginal; however, these factors improved the final model as well. 

Table 5 and Table 6 present the influence of BS in two models with the subsequent inclusion of groups of predictors. Table 5 shows odds ratios (ORs) in the model, including six variables grouped according to demographic (gender, age, family structure) and individual factors (BS, SSE, SA). Table 6 shows ORs for the model which includes all 11 potential predictors, extended by socioeconomic status indices (FA and FSS) as well as social factors (family and peer support, connections to others). In both models, BS has proven to be an important factor in reducing the risk of becoming a cybervictim or a cyberbully-victim. Boys were more at risk of becoming bullies and bully-victims than girls, and 13- and 15-year-olds, compared to the 17-year-olds, were found to be at risk of experiencing violence across all three forms of cyberbullying. Being from a family without parents, compared to living in an intact family, increased the risk of becoming a bully-victim, and this factor was a significant predictor of reactive victimization of cyberbullying in both models. Moreover, in the extended model (Table 6), it is noted that adolescents growing up in a single-parent family compared to an intact family are at risk of being a perpetrator. In a less complex model (Table 5), the protective effect of better school achievements among bully-victims was observed, which disappeared when other factors were added to the more extended model (Table 6). In perpetrators, the protective role of this factor remained in both models. In a simpler model, a protective effect of SSE on the risk of becoming a bully-victim was found; however, this effect disappeared in an extended model. In the extended model (Table 6), FA is a significant factor, and adolescents of the more affluent families are at higher risk of becoming bully-victims. Higher FA proved to be a factor that increased the risk of engaging in cyberbullying as a perpetrator-victim; however, perceived family social status was not found to have a significant impact on involvement in any type of cyberbullying.

Family support appeared to be an important protective factor in all types of cyberbullying involvement, and peer support turned out to be protective from cybervictimization. Strong support from the family has proven to have a significant protective effect on involvement in all three forms of cyberbullying and feeling strong support from peers contributes significantly to reducing the risk of being cyberbullied.

Feeling strong connections to others was a factor reducing the risk of becoming a bully or a bully-victim but increased the risk of becoming only a cybervictim.

Figure 1 presents the changes in the OR indicators for the BS variable in subsequent models after the inclusion of groups of 11 potential predictors of cyberbullying involvement. As the BS was taken as a continuous variable OR illustrated how much the risk of being a bully, a victim or a bully-victim was reduced by one point of BS improvement. In the model constructed in this way, BS was a protective factor, and the OR values were smaller than 1, the lower the OR, the greater the protective effect. When the perpetrators were compared with adolescents without any cyberbullying experience, BS proved to be important only in three models that include sociodemographic, other individual, and socioeconomic factors. In the case of victims, the protective effect of BS was maintained in all five models presented in Figure 1, although it gradually weakened. OR coefficients increased from 0.927 in the model containing only sociodemographic variables to 0.936 in the full model with 11 predictors. A relatively large change occurred after the introduction of variables describing social support (family and peer support). When the group of both perpetrator-victims of cyberbullying was analyzed, the BS effect was much greater than for only perpetrators, and only slightly smaller compared to models for only victims. In bully-victims in all models, this effect was statistically significant, and the OR indicators for BS increased from 0.935 to 0.955. Similarly, the largest increase was due to the addition of social support. The results prove that social support most notably moderates the effect of BS on cyberbullying involvement. Connections to others is an important independent predictor of these cyberbullying experiences, but this factor moderated to a lesser extent the BS effect in victims and bully-victims, and to a relatively greater extent in perpetrators.

### 3.5. Gender Specific Models

When comparing two gender-specific extended models, the model for girls showed better parameters of fit. The results of the log-likelihood test showed that the R-sq in the model for girls was higher (Nagelkerke R-sq = 0.127) than in the model for boys (Nagelkerke R-sq = 0.101). When comparing the risk of cyberbullying involvement between girls and boys, differences in predictions were found. In Table A3 and Table A4, provided as supplementary electronic material, the results of the multinomial logistic regression of the gender-specific extended models with 10 factors are presented.

The influence of body image on cyberbullying experiences as victims and as bully-victims has been confirmed in both genders, as it has been found in the general model for the entire study sample (for victims OR = 0.936; CI:0.918–0.953; for bully-victims OR = 0.955; CI:0.937–0.972). A slightly stronger protective effect of higher body BS in cyberbullying involvement has been observed in boys compared to girls as victims (boys: OR = 0.928; CI:0.898–0.960; in girls OR = 0.939; CI:0.918–0.961) and as bully-victims (boys: OR = 0.947; CI:0.923–0.972, in girls OR = 0.959; CI:0.935–0.985).

Although the model estimated for the full sample showed a significant protective effect of SSE on being a perpetrator of cyberbullying, the relationship of this factor to any type of involvement in cyberbullying was not confirmed in gender-specific models. Higher school achievements remained a significant predictor only for girls as lowering the risk of involvement as perpetrators and bully-victims. Family support remained a significant factor reducing involvement in perpetration or victimization for girls and in bully-victimization for boys. Peer support remained a significant factor of cybervictimization only for girls. A relationship between connections to others with all types of cyberbullying involvement observed in the model for the full sample has been proved for boys. Connections to others lowered the risk of becoming a bully or bully-victim and increased the risk of becoming a victim of cyberbullying among boys. For girls, a protective function of connections to others was related with the risk of being only a perpetrator or a bully-victim.

## 4. Discussion

The intense increase in adolescents’ virtual world presence, especially the use of social media, contributes to their increasing exposure to idealized body images and also results in increased exposure to appearance-related messages. It can cause body image concerns and diminish their body satisfaction [1]. This study draws attention to the role of BS in relation to other psychosocial determinants of adolescent’s cyberbullying involvement. This paper is based partially on a complex social-ecological theory, which includes individual and social determinants of cyberviolence [28,50]. Based on the literature review, 11 psychosocial determinants were selected and grouped into three substantive areas: sociodemographic, individual and social factors. The application of multinomial logistic regression enabled us to assess these determinants and estimate the chances of finding oneself in each of these three types, taking persons who were not involved in cyberbullying as a reference group. A number of alternative determinant models have been proposed by including categories of factors one by one, and in this manner, the moderation effect of BS was indirectly examined. Which factors enhance and which undermine the role of BS in adolescents’ cyberbullying has been demonstrated. Our findings have proved that BS reduces significantly the risk of cyberbullying involvement in all three groups, in bullies, victims and bully-victims. The relationship was more pronounced in victims and bully-victims, and a slightly stronger protective effect of body satisfaction has been observed in boys. In bullies, BS proved to be important in models that included sociodemographic, other individual and socioeconomic factors. In victims the protective effect of BS was maintained in all models, although it gradually weakened. In bully-victims, the BS effect was much greater than for only perpetrators, and only slightly smaller compared to models for only victims. The results proved that social support (family and peer support) most notably moderates the effect of BS on cyberbullying involvement.

Although an overview of the literature based on the results of the previous HBSC surveys indicates several studies connected to our own [18,50,51,52], none of them discussed the role of BS in relation to other psychosocial determinants in cyberbullying involvement in such a comprehensive manner. These studies were more likely to address the issue of cybervictimization [18,52,53,54]. A group of cyberbullies-victims was distinguished only sporadically, as it appeared more frequently in works related to ‘traditional’ bullying [55,56,57]. Our study stands out among other publications due to the fact that it underlines the moderating role of BS among a number of other psychosocial determinants of adolescents’ cyberbullying involvement.

The last international report of the HSBC shows results regarding cybervictimization and cyberperpetration only and does not consist of both cyberperpetration and victimization as a third type of cyberbullying involvement [34]. The distinction between the types of cyberbullying involvement as a bully, victim and a bully-victim was made accordingly to studies which examine the psychosocial factors that contribute to adolescents’ cyberviolence [9,10,50]. Our findings show that 25% of the adolescents in Poland were somehow involved in cyberbullying. While one in ten of respondents admitted to being both a bully and a victim, this group was identified as requiring special attention. The frequency of being a cybervictim only or a cyberperpetrator was similar, namely 7.8% and 7.1%, respectively. In our study, the bully-victims group emerged as more common (10.1%), and boys were more likely to be in this group than girls (12.8% vs 7.6%). In other studies, the prevalence of cybervictimization and perpetration are diverse, depending on the specific age range of the adolescents examined [10,58].

In our study, gender appeared to be significant factor, and boys were more at risk of perpetration and bullying-victimization. Other studies show various results regarding gender differences, and in the study of Li [59] boys were more likely to be cyberbullies than girls, and in the research of Mishna et al. [10], girls were more involved as cyberbully-victims. Many studies have shown that adolescents, especially girls, who are dissatisfied with their body more frequently experience cyberbullying directed at their appearance, so their body dissatisfaction strongly predicts the probability of being a cybervictim [7,17,18]. On the other hand, studies show that cybervictimization predicts lower body related self-esteem, especially in girls [60], so the relationship could be bidirectional.

In our study, family support was found to have the strongest protective effect on cyberbullying involvement of all three types, and peer support contributed additionally to reducing the risk of cybervictimization. Other studies have proved the protective role of family only in victimization and the bullying-victimization [20,31,32,61]. The systematic review of dozens of longitudinal studies published between 2007 and 2017 confirmed that family and peer support, and attachment decrease the risk of becoming a bully and a bully-victim, although the evidence on the causal relationships was scarce [62]. Connections to others decreased the risk of bullying and bullying-victimization, but increased the risk of victimization in the study group. Our results support the findings of other studies, which emphasize the importance of empathy, moral engagement and commitment to respecting others in cyberbullying prevention [26,27].

Our study shows that adolescents’ cyberbullying involvement as a victim and bully-victim can be explained to a relatively large extent by BS, and in these groups the association with BS was most clearly evident. High BS decreases the risk of victimization and both perpetration-victimization. Guo et al. [50] showed that body image was unique for distinctive types of cyberbullying involvement, and this factor appeared to be a stronger predictor that distinguished victims and bully-victims.

In our study, the strong association of BS with cybervictimization and bullying-victimization was observed, and its strong effect persisted even after adding a set of other psychosocial determinants to the models analyzed. Regarding cyberbullying perpetration, the association with body image was much weaker. These findings are in line with Brixval et al.’s study [16], where low BS in adolescents was found to be associated with more frequently experiencing weight-related bullying both for boys and girls. The explanation for this relationship can be sought in the concept of the appearance-based rejection sensitivity in adolescence. It is defined as a personality processing system characterized by anxiousness and expectations of rejection based on physical attractiveness [63]. Taking into consideration that physical appearance is highly valued by teenagers, body dissatisfaction predicts the probability not only of victimization [20], but also bullying-victimization, what has been shown in our study.

Connections to others was a variable which moderates to a greater extent the BS effect in perpetrators and to a lesser extent in victims and bully-victims. This could be explained by the definition of this specific domain of adolescents’ spiritual health, where meaningful connections to others are strongly influenced by the connections young people have with their social world around them, and first of all the quality of friendships and the relationships with their peers [64]. Family and peer support was found to partially moderate the relationship between BS and all types of cyberbullying involvement. This is with line with other studies, which showed that social support mediates the relationship between cyberbullying and perceptions of feeling too fat, for both boys and girls [17,18]. Our results also showed, both in boys and girls, that the cyberbullying involvement as a victim and as a bully-victim has been explained to a great extent by BS, with a slightly stronger protective effect among boys compared to girls. Other studies have pointed out the gender difference too, and have shown that cyberbullied girls were three times more likely than boys to report that their body is too fat, but, among victims, more girls demonstrated lower body-related self -esteem compared to boys [17,18].

This cross-sectional study indicates relationships between BS and types of cyberbullying involvement and does not provide grounds for conclusions based on cause and effect. It may be considered that body dissatisfaction may be caused by body and appearance-related cyberbullying, but this relationship might be bidirectional, and low body related self-esteem might cause perpetration or victimization [8,60]. Longitudinal studies are needed to clarify whether body satisfaction may be antecedents or outcomes of cyberbullying involvement. In our analysis, only two items available in the protocol about cyberbullying have been used, and dichotomous division has been conducted, into those who were involved in cyberbullying and those who were not, and then four clusters were defined based on respondents’ cyberbullying status. Collection and analysis of the data regarding frequency, forms, and circumstances related to cyberbullying among adolescents would be worthwhile to carry out, as this would allow to estimate the dose effect, which is the exposure and severity of cyberviolence [65].

Nevertheless, it seems that a number of other factors in this study offset the above limitations. One of them is the large representative study sample, consisting of 5817 adolescents. A standardized questionnaire was used, and the scales were previously verified in other studies. Another advantage is the assessment of relationship between body satisfaction and cyberbullying involvement among adolescents from Poland—a country which tops the HBSC countries rankings in both cases with very unfavorable indicators, especially in poor body image among girls.

## 5. Conclusions

As electronic technology in communication and social media use is likely to continue to increase, cyberbullying aimed at the body and appearance will continue to be an important problem for adolescents’ wellbeing. Our results suggests that there is variability between psychosocial characteristics of cyberbullies, cybervictims and cyberbully-victims, and body satisfaction is one of the important determinants among other psychosocial predictors.

Our results suggest that policies and programs should consider body related self-esteem in cyberbullying prevention, especially aimed at victimization and bully-victimization. Efforts to address protective cybertechnologies through which the cyberbullying is perpetrated, need to be specific to different groups involved in cyberbullying. The strengthening of body satisfaction may be an effective measure to prevent cyberbullying among adolescents, besides improving family and peer support, developing connections to others, and increasing school performance satisfaction. The body image reinforcement should be considered as an important preventive measure of cybervictimization and cyberbullying-victimization, especially in boys.

## Figures and Tables

**Figure 1 ijerph-19-01292-f001:**
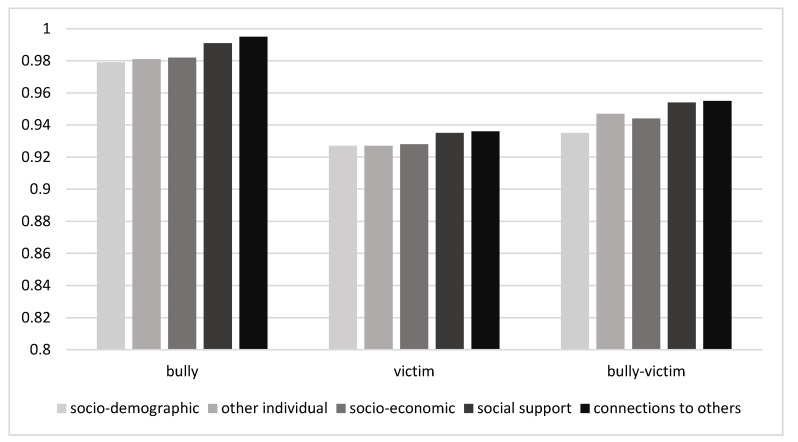
Change in odds ratio (OR) for the body satisfaction in multinomial logistic regression models after including blocks of other variables.

**Table 1 ijerph-19-01292-t001:** Cyberbullying involvement by the sociodemographic characteristics.

Variables	N (%)	Cyberbullying Involvement —At Least 2 Times in the Last Two Months (%)				*p*
		Not Involved	Only Bully	Only Victim	Bully and Victim	
Total (%)		75.0	7.1	7.8	10.1	
Gender						
Boys	2776 (47.6)	72.3	8.6	6.3	12.8	0.000
Girls	3061 (52.4)	77.4	5.7	9.3	7.6	
Age						
13 years	2144 (36.7)	73.9	7.3	9.3	9.5	
15 years	1993 (34.1)	72.2	7.8	8.3	11.8	0.000
17 years	1700 (29.1)	79.6	6.0	5.5	8.8	
Family structure						
Intact	4390 (75.2)	75.8	7.3	7.8	9.1	
Single parent	961 (16.5)	74.6	5.5	7.4	12.4	0.000
Step-parent	307 (5.3)	69.0	8.5	9.8	12.7	
Without parents	180 (3.1)	67.8	7.3	7.9	16.9	
Family affluence						
Low (0–5)	946 (16.5)	75.3	6.9	8.8	9.0	
Average (6–9)	3426 (59.8)	75.7	6.9	7.1	10.2	0.198
High (10–13)	1354 (23.6)	72.8	7.7	8.9	10.6	
Family social status						
Low (0–5)	1101 (19.2)	72.9	6.7	8.7	11.8	
Average (6–8)	3276 (57.1)	75.0	7.1	7.7	10.3	0.117
High (9–10)	1362 (23.7)	76.6	7.7	7.3	8.5	

**Table 2 ijerph-19-01292-t002:** Cyberbullying involvement by body satisfaction and other individual factors.

Variables	N (%)	Cyberbullying Involvement —At Least Two Times in the Last Two Months (%)				*p*
		Not Involved	Only Bully	Only Victim	Bully and Victim	
Body Satisfaction						
Not satisfied (0–12)	1511 (26.5)	68.5	6.4	12.2	12.9	0.000
Average (13–20)	2828 (49.6)	76.4	6.9	6.9	9.7	
Satisfied (21–24)	1366 (23.9)	79.9	7.7	5.0	7.4	
Social self-efficacy						
Low (0–16)	1381 (24.4)	72.3	6.9	8.4	12.4	
Average (17–25)	3044 (53.8)	75.2	6.9	7.6	10.3	0.000
High (26–32)	1229 (21.7)	78.0	7.6	7.7	6.7	
School achievements						
Low (0–4)	1117 (19.3)	69.8	8.6	8.4	13.1	
Average (5–7)	3082 (53.2)	74.9	7.0	8.1	10.0	0.000
Very good (8–10)	1590 (27.5)	78.7	6.2	7.0	8.1	

**Table 3 ijerph-19-01292-t003:** Cyberbullying involvement by the social support and connections to others.

Variables	N (%)	Cyberbullying Involvement —At Least Two Times in the Last Two Months (%)				*p*
		Not Involved	Only Bully	Only Victim	Bully and Victim	
Family support						
Weak (0–12)	1497 (25.9)	67.6	8.2	10.0	14.2	
Average (13–22)	2825 (49.0)	75.3	7.4	7.6	9.7	0.000
Strong (23–24)	1447 (25.1)	82.2	5.4	6.0	6.4	
Peer support						
Weak (0–9)	1497 (25.8)	69.0	8.0	10.1	12.8	
Average (10–19)	2957 (51.1)	76.5	6.3	7.2	10.0	0.000
Strong (20–24)	1338 (23.1)	78.3	7.6	6.8	7.3	
Connections to others						
Weak (0–8)	1108 (19.1)	62.4	11.3	6.3	19.9	
Average (9–10)	2302 (39.7)	74.9	8.0	7.5	9.6	0.000
Strong (11–12)	2393 (41.2)	81.0	4.2	8.8	6.0	

**Table 4 ijerph-19-01292-t004:** The correlations of psychosocial determinants of cyberbullying involvement.

Variables	Mean (SD)	1	2	3	4	5	6	7	8
1. Body satisfaction (0–24)	15.93 (5.60)	1.000	0.237 ***	0.155 ***	0.040 *	0.203 ***	0.304 ***	0.166 ***	0.114 ***
2. Social self-efficacy (0–32)	20.62(6.07)		1.000	0.140 ***	0.131 ***	0.192 ***	0.285 ***	0.457 ***	0.230 ***
3. School achievements (0–10)	6.19(2.07)			1.000	0.151 ***	0.227 ***	0.176 ***	0.140 ***	0.136 ***
4. Family affluence (0–13)	7.78 (2.32)				1.000	0.321 ***	0.091 ***	0.063 ***	0.004
5. Family social status (0–10)	7.15 (1.82)					1.000	0.341 ***	0.144 ***	0.080 ***
6. Family support (0–24)	16.72(6.44)						1.000	0.295 ***	0.270 ***
7. Peer support (0–24)	13.94(6.47)							1.000	0.274 ***
8. Connections to others (0–12)	9.69(2.34)								1.000

* *p* < 0.05; *** *p* < 0.001.

**Table 5 ijerph-19-01292-t005:** Risk of cyberbullying involvement by sociodemographic and individual factors, including body satisfaction, estimated by multinomial logistic regression.

Independent Variables	Only Bully			Only Victim			Bully-Victim		
	*p*	OR	95% CI	*p*	OR	95% CI	*p*	OR	95% CI
Constant	0.000			0.000			0.000		
Gender									
Boys	0.000	1.732	1.38–2.1637	0.292	0.890	0.717–1.105	0.000	2.079	1.714–2.521
Girls (ref.)		1.000			1.000			1.000	
Age									
13 years	0.042	1.325	1.010–1.739	0.000	2.161	1.651–2.828	0.032	1.293	1.022–1.637
15 years	0.003	1.502	1.147–1.968	0.000	1.810	1.372–2.389	0.000	1.564	1.243–1.970
17 years (ref.)		1.000			1.000			1.000	
Family structure									
Single parent	0.061	1.737	0.535–1.014	0.313	0.863	0.649–1.149	0.129	1.201	0.948–1.522
Step-parent	0.150	1.371	0.892–2.105	0.347	1.226	0.802–1.875	0.062	1.440	0.982–2.110
Without parents	0.507	1.222	0.676–2.206	0.845	0.938	0.496–1.774	0.007	1.852	1.180–2.905
Intact (ref.)		1.000			1.000			1.000	
Body satisfaction	0.068	0.981	0.961–1.001	0.000	0.927	0.910–0.944	0.000	0.942	0.926–0.958
Social self-efficacy	0.237	1.011	0.993–1.030	0.713	1.003	0.986–1.021	0.032	0.984	0.969–0.999
School achievements	0.003	0.922	0.875–0.972	0.442	0.980	0.932–1.031	0.003	0.935	0.894–0.978

Note. Data are odds ratio’s (OR) and 95% confidence intervals (CI). ref. = reference category.

**Table 6 ijerph-19-01292-t006:** Risk of cyberbullying involvement by all 11 psychosocial factors estimated by multinomial logistic regression.

Independent Variables	Only Bully			Only Victim			Bully-Victim		
	*p*	OR	95% CI	*p*	OR	95% CI	*p*	OR	95% CI
Constant	0.000			0.000			0.758		
Gender									
Boys	0.001	1.509	1.197–1.904	0.688	0.955	0.763–1.196	0.000	1.849	1.508–2.268
Girls (ref.)		1.000			1.000			1.000	
Age									
13 years	0.032	1.366	1.028–1.816	0.000	2.167	1.641–2.862	0.004	1.450	1.129–1.862
15 years	0.002	1.537	1.164–2.030	0.000	1.682	1.267–2.233	0.000	1.649	1.295–2.099
17 years (ref.)		1.000			1.000			1.000	
Family structure									
Single parent	0.021	0.665	0.471–0.939	0.239	0.834	0.617–1.128	0.111	1.226	0.954–1.576
Step-parent	0.138	1.391	0.900–2.150	0.308	1.250	0.814–1.921	0.109	1.386	0.930–2.067
Without parents	0.448	1.265	0.689–2.324	0.997	0.999	0.524–1.902	0.030	1.721	1.053–2.811
Intact (ref.)		1.000			1.000			1.000	
Body satisfaction	0.662	0.995	0.973–1.017	0.000	0.936	0.918–0.953	0.000	0.955	0.937–0.972
Social self-efficacy	0.038	1.023	1.001–1.045	0.065	1.019	0.999–1.040	0.418	1.007	0.990–1.025
School achievements	0.018	0.935	0.885–0.988	0.885	0.996	0.944–1.051	0.178	0.968	0.922–1.015
Family affluence	0.302	1.027	0.976–1.080	0.534	1.015	0.968–1.065	0.008	1.061	1.015–1.109
Family social position	0.071	1.066	0.995–1.143	0.845	1.006	0.944–1.051	0.065	0.947	0.894–1.003
Family support	0.013	0.976	0.957–0.995	0.000	0.968	0.951–0.986	0.003	0.975	0.959–0.991
Peer support	0.204	1.013	0.993–1.034	0.010	0.976	0.958–0.994	0.753	0.997	0.980–1.015
Connections to others	0.000	0.823	0.788–0.859	0.023	1.065	1.009–1.125	0.000	0.832	0.802–0.864

Note. Data are odds ratio’s (OR) and 95% confidence intervals (CI). ref. = reference category.

## Data Availability

The data presented in this study are available on request from the corresponding author. The data are not publicly available due to internal HBSC data access policy. Data access to previous HBSC rounds is provided by the HBSC Data Management Centre—Department of Health Promotion and Development, University of Bergen (https://www.uib.no/en/hbscdata (accessed on 21 December 2021)).

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
