# Peer review of "Psychosocial Determinants of Adolescents’ Cyberbullying Involvement—The Role of Body Satisfaction"

_ijerph, 2022, doi:10.3390/ijerph19031292_

Round 1
Reviewer 1 Report
I would like to thank the editor and the authors for the opportunity to review this manuscript entitled: Psychosocial determinants of adolescents’ cyberbullying involvement. The role of body satisfaction. Generally, the article is well written and addresses the current and under researched topic of cyberbullying involvement.
Theoretical part is appropriately structured, with adequate background. The hypotheses of the research were clear.
In general, subject selection is clear, however, I suggest adding relevant details on how the questionnaire (scales) was adapted (1).
In terms of analysis, I suggest adding information on how the authors checked the assumptions (e.g. no multi-collinearity) to ensure that models were valid (2).
The results are presented in an appropriate way. Tables and figures are relevant and clearly presented.
The results are discussed comprehensively but however to my mind this section should be reformulated and placed (3) more clearly into context with given hypotheses (so that readers can see how the results answer the hypotheses of the study/and conclusions supported by references). Furthermore, part of the theoretical discussion at the first paragraph could fit to the context of presentation of aims/hypotheses of the study.
I hope these comments are useful.
Author Response
Response to Reviewer 1 Comments
Thank you for your kind and constructive comments and feedback. Kindly find our response to the comments below.
Point 1: In general, subject selection is clear, however, I suggest adding relevant details on how the questionnaire (scales) was adapted
Response 1: In Materials and Methods in 2.1. we added the information translation and back translation of the scales included in the questionnaire was adopted. The translation procedure of the questionnaire followed an international HBSC survey protocol (Inchley et al. 2018). All questions were translated from the original English version into the Polish. Thereafter the Polish translation was back translated into English by an independent translator and submitted to the HBSC Translation Hub. The translation was reviewed, accepted or modified according to the reviewer's comments.
Point 2: In terms of analysis, I suggest adding information on how the authors checked the assumptions (e.g. no multi-collinearity) to ensure that models were valid
Response 2: In Results in 3.3. we added information that we checked the assumptions regarding multicollinearity, to ensure the regression-type analysis, that the models were valid (Vatcheva et al. 2016; Shrestha, 2020). The values of the correlation coefficients between independent variables used in our study were quite low (|rho| < 0.7). According to general assumptions it allowed the regression analyses to be performed, and to estimate the models.
Point 3: The results are discussed comprehensively but however to my mind this section should be reformulated and placed (3) more clearly into context with given hypotheses (so that readers can see how the results answer the hypotheses of the study/and conclusions supported by references). Furthermore, part of the theoretical discussion at the first paragraph could fit to the context of presentation of aims/hypotheses of the study.
Response 3: The part of the Discussion at the first paragraph has been completed to better fit to the context of the aim and hypothesis of the study. We added that our findings have proved that BS reduces significantly the risk of cyberbullying involvement in all three groups, in bullies, victims and bully-victims. The relationship was more pronounced in victims and bully-victims, and a slightly stronger protective effect of body satisfaction has been observed in boys. In bullies BS proved to be important in models that included socio-demographic, other individual and socio-economic factors. In victims the protective effect of BS was maintained in all models, although it gradually weakened. In bully-victims the BS effect was much greater than for only perpetrators, and only slightly smaller compared to models for only victims. The results proved that social support (family and peer support) most notably moderates the effect of BS on cyberbullying involvement.

Reviewer 2 Report
Using a large sample and a standardized questionnaire, the study conducted a reliable design to explore and assess the relationship between body image and adolescents’ cyberbullying involvement as bully, victim and bully-victim in the context of other psychosocial predictors. I raise several comments on this article, hoping to make the research presented more precisely.
- While measuring family structure and socio-economic status, Cronbach’s alpha was less than 0.7 and PCA was only 33%. Though it is an accepted tool implemented by the HBSC research network, there may be a little difference while using it among countries. I suggest the authors try to delete one or two of those items to increase the discrimination score. Moreover, why did the authors demonstrate that a continuous family affluence (FA) variable was used in analyses?
- From Table 1 to Table 3, the total number the study included is 5817. However, there are different sum in several factors. In the family structure, the total number is even more than 5817. The authors need to clarify the numeric. Even if a missing occurs, a detailed statement is needed.
- In Table 4, several correlations were observed being strong significantly. As they were all the independent variables, the collinear among those variables seem to be concerned.
- In Table 5 & the appendix, the p-value usually is presented with <0.001, instead of 0.000.
- In Table 6, a typo (0.471-.0939) was found in 95% CI of single parents in Only bully. So did "Family social position" in Table A4.
- As the study tries to explore the moderation of factors on the relationship between body satisfaction and cyberbullying involvement, a structural equation model is suggested to analyze the relationships precisely.
Author Response
Response to Reviewer 2 Comments
Thank you for your kind and constructive comments, feedback and remarks. Kindly find our response to the comments below.
Point 1: While measuring family structure and socio-economic status, Cronbach’s alpha was less than 0.7 and PCA was only 33%. Though it is an accepted tool implemented by the HBSC research network, there may be a little difference while using it among countries. I suggest the authors try to delete one or two of those items to increase the discrimination score. Moreover, why did the authors demonstrate that a continuous family affluence (FA) variable was used in analyses?
Response 1: In Materials and Methods, Section 2.2.2. we explained that the socio-economic status was measured by the Family Affluence Scale (FAS III), and in correlation analysis family affluence variable was used as continuous variable. Even if Cronbach’s alpha for FAS III was less than 0.7 with PCA only 33.5%, this tool is recommended for use in HBSC study. The six-item scale was estimated with Samejima's graded response model, and tested for differential item functioning by country (also Poland) under the last FAS scale validation study (Torsheim et al, 2016). This study the test-retest reliability for Poland was r=0.91 and the FAS scale correlated with the family income reported by parents with the Eta2 close to 0.30.
Point 2: From Table 1 to Table 3, the total number the study included is 5817. However, there are different sum in several factors. In the family structure, the total number is even more than 5817. The authors need to clarify the numeric. Even if a missing occurs, a detailed statement is needed.
Response 2: In Results, section 3.1. we added information that the total number of the study group included 5817 adolescents. However there are different sum in variables due to missing data. At the top of the Table 1, Table 2 and Table 3, we deleted the numbers to clarify the numeric. Moreover we noticed that we missed the Table 2 entitled “Cyberbullying involvement by the body satisfaction and other individual factors”. We corrected this mistake.
Point 3: In Table 4, several correlations were observed being strong significantly. As they were all the independent variables, the collinear among those variables seem to be concerned.
Response 3: In the Results in the section 3.3. we added the information that we checked the assumptions regarding multicollinearity, to ensure the regression-type analysis, that the models were valid (Vatcheva et al. 2016; Shrestha, 2020). The values of the correlation coefficients between independent variables used in our study were quite low (|rho| < 0.7). According to general assumptions it allowed the regression analyses to be performed, and to estimate the models.
Point 4: In Table 5 & the appendix, the p-value usually is presented with <0.001, instead of 0.000.
Response 4: We corrected this.
Point 5: In Table 6, a typo (0.471-.0939) was found in 95% CI of single parents in Only bully. So did "Family social position" in Table A4.
Response 5: We corrected this.
Point 6: As the study tries to explore the moderation of factors on the relationship between body satisfaction and cyberbullying involvement, a structural equation model is suggested to analyze the relationships precisely.
Response 6: We appreciate the suggestion to use a structural equation model to explore the moderation effect of the body satisfaction and cyberbullying involvement. We chose this method of analysis to be able to compare different participation in cyberbullying as a bully, a victim, and a bully-victim. Nevertheless, in the future, it is certainly worth considering using SEM to explore the relationship between body satisfaction and cyberbullying among adolescents.

Reviewer 3 Report
Good job! Your article represents a good contribution, especially because it includes the study of contextual variables that allow us to understand the participation of students in cyberbullying.
Some important observations would be:
- In the summary, there is an error in the repetition of content. It has information splicing. The keywords are repeated.
- There are major errors in the results. Table 2 shows the same results as table 1 and not what corresponds to individual factors measured.
- Table 4 does not explicitly show information on the level of significance of the correlations. I think it is important to include this information, even when it is described in the text the table should include it.
Author Response
Response to Reviewer 3 Comments
Thank you for your kind and constructive comments, feedback and remarks. Kindly find our response to the comments below.
Point 1: In the summary, there is an error in the repetition of content. It has information splicing. The keywords are repeated.
Response 1: We corrected this.
Point 2: There are major errors in the results. Table 2 shows the same results as table 1 and not what corresponds to individual factors measured.
Response 2: We corrected this mistake.
Point 3: Table 4 does not explicitly show information on the level of significance of the correlations. I think it is important to include this information, even when it is described in the text the table should include it.
Response 3: We completed the missing p values by signing the level of the significance, and explanation in the footnote of the Table 4.
